**Subject Category:**
Biology (whole organism)

ecology

climate change, food caching, hoard-rot, long-term data, *Perisoreus canadensis*, seasonal interactions

**Author for correspondence:**
Alex O. Sutton
e-mail: asutto01@uoguelph.ca

# Autumn freeze-thaw events carry over to depress late-winter reproductive performance in Canada jays

Alex O. Sutton[1], Dan Strickland[2], Nikole E. Freeman[1], Amy E. M. Newman[1] and D. Ryan Norris[1]

[1]Department of Integrative Biology, University of Guelph, Guelph, Ontario, Canada N1G 2W1
[2]1063 Oxtongue Lake Road, Dwight, Ontario, Canada P0A 1H0

AOS, 0000-0002-0311-7883; DRN, 0000-0003-4874-1425

Evidence suggests that range-edge populations are highly vulnerable to the impacts of climate change, but few studies have examined the specific mechanisms that are driving observed declines. Species that store perishable food for extended periods of time may be particularly susceptible to environmental change because shifts in climatic conditions could accelerate the natural degradation of their cached food. Here, we use 40 years of breeding data from a marked population of Canada jays (*Perisoreus canadensis*) located at the southern edge of their range in Algonquin Provincial Park, Ontario, to examine whether climatic conditions prior to breeding carry over to influence reproductive performance. We found that multiple measures of Canada jay reproductive performance (brood size, nest success and nestling condition) in the late winter were negatively correlated with the number of freeze–thaw events the previous autumn. Our results suggest that freeze–thaw events have a significant detrimental impact on the quality and/or quantity of cached food available to Canada jays. Future increases in such events, caused by climate change, could pose a serious threat to Canada jays and other food caching species that store perishable foods for long periods of time.

## 1. Introduction

There is strong evidence to suggest that long-term changes in climate have affected a wide range of species around the world [1–3]. A major consequence of climate change is a poleward shift in the distribution of species [4,5], whereby populations living at lower latitudes of a species' distribution decline are eventually extirpated because they are most susceptible to

warming temperatures [6]. However, documentation of such range-edge population declines is rare [7,8] and even fewer specific mechanisms have been proposed to explain how changes in climate could be causing declines [3,6].

One group of animals that may be vulnerable to climate change are those that cache food, especially perishable food [9]. Food caching is a behavioural strategy in which an animal defers consumption of a food item and handles it in a way that deters other organisms from accessing stored food so that it can be retrieved during a period when demand for resources outweighs supply [10]. Once food is cached, it is exposed to environmental conditions that may reduce its quality over time [9,11]. The degree to which food degrades depends on both the type of food that is stored, as well as where and how long it is stored [9]. For example, foods such as nuts and seeds are probably more resistant to changes in environmental conditions compared to perishable foods such as meat, berries and fleshy fungi [9,10].

Canada jays (*Perisoreus canadensis*) are year-round residents of the boreal forest that cache a wide variety of perishable foods including arthropods, mushrooms, berries and vertebrate flesh (both carrion and prey) in the late summer and autumn [12]. Canada jays are arboreal scatter-hoarders and distribute food caches widely across their territories [12]. Foraging for food is not a conspicuous activity in winter, and Canada jays do not exploit boreal tree seed crops or have regular access to animal carcasses in otherwise seemingly foodless boreal winter conditions [12]. By contrast, food caching is very prominent in the summer and autumn (e.g. individuals can make as many as 1000 caches a day in Alaska; T. Waite 1991, unpublished data). This evidence strongly implies that large quantities of food are stored throughout a typical jay territory and almost certainly account for their observed high territorial fidelity and winter survival [12,13]. Cached food also supports female increases in body mass prior to clutch initiation [14] and some evidence suggests that breeding pairs also use stored food to feed nestlings, at least occasionally [15]. Critically, however, and notwithstanding the high perishability of food cached by Canada jays, food must survive in sufficient quantity or quality from the time of storage in summer or autumn to the following breeding season if it is to explain the high winter survival and late-winter breeding performance in this species.

Until the onset of consistent sub-freezing winter temperatures, the temperature-dependent degradation of perishable food items stored in the autumn is expected to be an important determinant of food quality [16]. Waite & Strickland [17] documented a decline in our southern range-edge population of Canada Jays and proposed that warmer autumn temperatures might be accelerating declines in the quality of cached perishable food items, which then carry over to influence reproductive performance. Using 26 years of data from Algonquin Provincial Park, Ontario (APP), they provided some support for their 'hoard-rot' hypothesis by showing that both clutch size and the timing of breeding were negatively correlated with mean autumn temperatures. However, the support for both models was relatively weak and there was no evidence that autumn temperature influenced brood size, one of the key indicators of reproductive performance.

One possible explanation for this relative lack of support is that other changing features of our study area's warming climate (i.e. beyond higher temperatures) may be contributing to declines in the quality or quantity of stored food, ostensibly resulting in the observed decline of our study population. Food scientists have identified a number of environmental variables, including temperature, and freeze–thaw events, that influence food quality and which may be relevant to wild populations [9]. Temperature affects the quality of perishable food primarily through its influence on microbial growth rates [18]. By contrast, freeze–thaw events act in three distinct ways to degrade perishable food. First, freezing and thawing of a food item cause the denaturation of proteins, which can have downstream consequences on nutritional quality [19–21]. Second, as water undergoes phase changes (i.e. liquid to solid and vice versa) it causes cell membranes to rupture and this breakage allows enhanced physical leakage during subsequent thaw periods. This process of the soluble contents of the cell leaving is known as 'drip-loss' [22–24]. Finally, 'drip-loss' also causes enhanced microbial degradation during thaw periods because nutrients are easily accessed by microbes both within a cell and in the area surrounding the damaged food item [25]. By extension, perishable food cached by Canada jays could be susceptible to degradation caused by freeze–thaw events and thus they should be considered when evaluating the 'hoard-rot' hypothesis.

Our goal was to examine how climate variables linked to specific food degradation mechanisms might influence Canada jay reproductive performance. Using data on reproductive performance from an individually marked population of Canada jays in APP, we examined two hypotheses. The first, 'Warmer-Temperatures' hypothesis was that increasing mean temperatures during the autumn and pre-breeding periods lead to greater microbial degradation of perishable food caches that are used by jays during their late-winter breeding period. The second, 'Number of Freeze–Thaw Events'

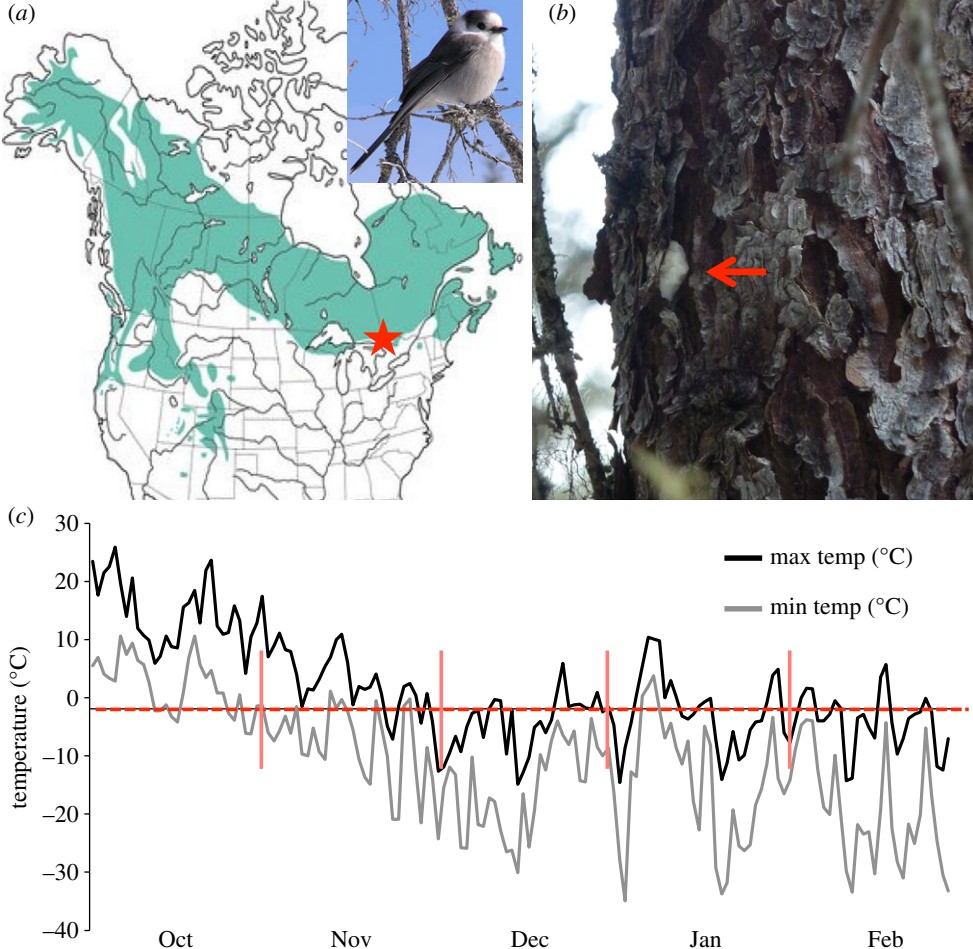

**Figure 1.** (*a*) Range of Canada jays (*Perisoreus canadensis*) across North America. The red star denotes the study area in Algonquin Provincial Park, Ontario. Inset is a Canada jay. (*b*) A Canada jay food cache in the autumn. Canada jays cache food boluses throughout their territories under arboreal lichens and bark flakes. The red arrow indicates the food bolus placed under a bark flake on a black spruce (*Picea mariana*). (*c*) An example of a temperature profile throughout the autumn caching period (October – December) and the pre-breeding period (January – February) from 2003. The dashed red line represents the initial freezing point (−1.9°C), which was used to calculate freeze – thaw events (see Methods). When maximum daily temperatures exceeded the initial freezing point and minimum temperatures were below the initial freezing point a freeze – thaw event was determined to have occurred.

hypothesis was that the number of freeze–thaw events that occurred during the autumn and pre-breeding periods would damage perishable cached food through both physical and microbial degradation, limiting the amount of food available during the late-winter breeding period. We tested predictions arising from these hypotheses by measuring brood size, nest success and nestling condition from 718 nests monitored between 1977 and 2016.

# 2. Methods

## 2.1. Study area and species

We conducted our study in Algonquin Provincial Park, Ontario, Canada (figure 1; APP; 45° N, 78° W), where Canada jays have been marked and monitored since 1964 [26–28]. Individuals within the study area maintain approximately 130 ha year-round territories and store food items under tree bark and arboreal lichens beginning in the late summer and early autumn [29]. In this analysis, we use data beginning in 1977 because few nests were found prior to this year and it also allowed us to include a higher proportion of known-age birds.

Each year ($N = 40$; 1977–2016) up to 25 nests (mean 18, range 9–25 including failed nests) in the study area were found and monitored throughout the breeding season (February–May). Nests were accessed when nestlings were approximately 11 days old, which is roughly halfway through the 22–24 day long nestling period. At the time when nests were accessed, we took morphometric measurements of young and determined brood size. All young were given a unique combination of coloured leg bands and a Canadian Wildlife Service aluminium band. Unmarked individuals that disperse into the study site, typically as juveniles, were captured during the autumn population census conducted in October and uniquely marked in the same way as nestlings.

## 2.2. Reproductive performance

We considered three metrics of reproductive performance:

(1) 'Brood size': the total number of nestlings in a nest at the time of banding (excluding nests that failed prior to nestlings being banded and measured). We did not use clutch size as a metric of reproductive performance because, logistically, we were unable to access nests multiple times throughout the reproductive period. However, we believe that, for the vast majority of nests, our brood size and clutch size were the same because there were very few cases where we observed additional eggs at the time of banding.

(2) 'Nest success': whether the first nest attempt of the year had young in the nest at the time they were banded and measured. We acknowledge that this may not be a true representation of fledging success because nestlings were banded and measured halfway through the nestling period. However, for a subset of nests that we monitored after banding, we observed few predation events, which suggests that the presence of nestlings at the time the nest was accessed was a good indicator that they will fledge from the nest.

(3) 'Nestling condition': mass given body size of a nestling at the time it is banded and measured in the nest (typically $\sim$ 11 days after hatching). Nestling condition was calculated following Derbyshire et al. [30] by first estimating body size using a principal component analysis (PCA; [31,32]) on a correlation matrix of tarsus, seventh primary and bill length measurements. This estimate was then regressed against mass using an asymptotic exponential model and residuals from this model were considered to be an estimate of body condition [30]. Negative residuals represent individuals that weighed less than expected for a given body size and therefore were assumed to be in below-average condition, while individuals with positive residuals weighed more than expected for a given body size and were assumed to be in above-average condition. We assumed that our condition estimate was a good estimate of body condition at the time of measurement, but have not formally investigated the downstream consequences of our condition estimates on survival. In total, 1263 nestlings from 718 nests were used in the analysis.

## 2.3. Historical weather data

Historical weather records (maximum, minimum and mean temperatures) for two periods of the annual cycle ('autumn': October–December and 'pre-breeding': January–February) were available from two weather stations. Weather data from 1977 to 2005 were collected from the Dwight weather station (45°23′ N  78°54′ W;  http://climate.weather.gc.ca/historical_data/search_historic_data_e.html),  15 km outside Algonquin's western boundary and data from 2004 to 2017 were collected at its East Gate (45°32′ N  78°16′ W;  https://weather.gc.ca/city/pages/on-29_metric_e.html),  located  in  the southeastern section of our study area. For the time period of overlap between the two weather stations (September 2004–December 2005), we compared temperatures from the two stations using reduced major axis (RMA) regression. RMA regression was used because there is potential measurement error associated with temperature (i.e. at each station; [33]). The regression equations generated were then used to transform data from the Dwight weather station. The relationship of all three temperature variables between the two weather stations appeared to vary seasonally, so we used two season-specific ('seasons' determined by the equinoxes) equations to adjust the data from the Dwight station (East Gate autumn temperature: $-0.949 + 0.997 \times$ Dwight autumn temperature, $R^2 = 0.96$, $p < 0.001$ and East Gate winter temperature: $-0.989 + 1.023 \times$ Dwight winter temperature, $R^2 = 0.91$, $p < 0.001$) were applied to the Dwight data.

## 2.4. Quantifying climate variables

Freeze–thaw events were calculated by estimating the 'initial freezing point' (IFP) of food types (meat, mushrooms and berries) that emulated food items consumed by Canada jays ([12]; see electronic supplementary material, table S1). IFP, the point at which ice crystal formation begins, is required to predict both physical and microbial properties of food [34]. We decided to use the mean of the initial freezing point of meat ($-1.9°C$) as the value to calculate freeze–thaw events from the historical climate records because this was the lowest value among the food types we surveyed from the literature (electronic supplementary material, table S2) and, therefore, a conservative representation of the initial freezing point for food consumed by jays (i.e. produced the fewest freeze–thaw events). We considered a freeze–thaw event to occur when daily maximum temperatures and daily minimum temperature fluctuated above and below the IFP. For example, if minimum temperatures were below the IFP on consecutive days and maximum temperature rose above IFP on one of these days, then we considered this to be a freeze–thaw event (figure 1). The total number of freeze–thaw events was then summed for both autumn (October–December) and pre-breeding (January–February) periods. Mean daily temperatures were averaged during the autumn and pre-breeding periods to calculate a mean temperature for each period.

## 2.5. Statistical analysis

To understand how the frequency of freeze–thaw events and mean temperature during the autumn and pre-breeding periods influenced reproductive performance, we constructed a series of generalized mixed effect models for each response variable (brood size: Poisson distribution; nest success: binomial distribution; nestling condition: Gaussian distribution). All models predicting brood size excluded failed nests (brood size = 0) because we were interested in variation in brood size of nests that had young at the time of banding. For each model series, we considered four climate variables: the frequency of freeze–thaw events during the autumn (Freeze-Autumn), the frequency of freeze–thaw events during the pre-breeding period (Freeze-Pre), mean temperature during the autumn (Temp-Autumn) and mean temperature during the pre-breeding period (Temp-Pre). Each series of models, including the base model without climate variables, included the following fixed effects: supplementation (whether individuals on specific territories were regularly fed by park visitors; see Derbyshire *et al.* [30]), male age, female age and first egg date (the date at which females began incubating). Each fixed effect was included based on previous evidence suggesting that it influences reproductive performance (supplementation [30]; male age [28]; female age [29,15]; lay date [35]). For the nestling condition model, brood size was also included as a fixed effect. All models also included year and male and female identity as random effects because many individuals bred in multiple years. For each response variable, we constructed models with single climate variables as well as all possible combination of climate variables in addition to the fixed effects in the base model. We also included an interaction between first egg date and freeze–thaw events or temperature because we were interested in examining whether the possible negative effects of these climate variables were stronger for pairs that nested later in the season given that (i) there is evidence that later nesting birds have lower reproductive success [28,30] and (ii) later nesting birds would retrieve caches that have been stored for longer periods of time to feed their young compared to earlier nesting birds.

Akaike's information criterion corrected for small sample sizes (AICc) was then used to rank competing models [36]. All models within $\Delta AICc \leq 2$ were considered as competing to describe variation in reproductive performance [35] and Akaike weights provided the cumulative support for a model given all competing models. All statistical tests and calculations were performed in R v. 3.3.2 [37] using the lme4 [38] and AICcmodavg [39] packages. Visreg [40] was used to visualize regression lines and confidence intervals for each regression line were estimated using bootpredictlme4 [41].

Fixed effects in each model predicting brood size, nest success or nestling condition were not highly correlated ($r < 0.3$).

## 3. Results

### 3.1. Description of climate variables and reproductive performance

Both mean temperature and frequency of freeze–thaw events varied considerably over the course of the study. Mean temperature in the autumn ranged from $-4.4°C$ to $2.4°C$ (mean $\pm$ s.d.; $-0.6°C \pm 1.5$), while

**Table 1.** Model comparison to explain variation in brood size of Canada jays using Akaike's information criterion for small sample sizes (AICc). The base model included level of supplementation, male age, female age and lay date but no climate variables. All other models also included these fixed effects. Climate variables added in each model are listed in addition to the total number of parameters in a model (K), AICc and ΔAICc scores and AICc weight. Freeze = frequency of freeze–thaw events, Temp = mean temperature, Autumn = autumn caching period (October–November), Pre = Pre-breeding period (January–February).

| Model predicting brood size | K | AICc | ΔAICc | AICc weight |
|---|---|---|---|---|
| Freeze-Autumn + Temp-Pre + Freeze-Autumn × Lay Date | 11 | 2178.66 | 0 | 0.23 |
| Freeze-Autumn + Temp-Pre | 10 | 2178.76 | 0.10 | 0.22 |
| Freeze-Autumn + Temp-Pre + Freeze-Pre | 11 | 2178.86 | 1.20 | 0.13 |
| Freeze-Autumn + Temp-Autumn + Temp-Pre | 11 | 2180.35 | 1.69 | 0.10 |
| Freeze-Autumn | 8 | 2180.78 | 2.11 | 0.08 |
| Freeze-Autumn + Temp-Autumn | 9 | 2181.52 | 2.85 | 0.06 |
| Freeze-Autumn + Temp-Autumn + Temp-Pre + Freeze-Pre | 12 | 2181.58 | 2.92 | 0.05 |
| Temp-Pre | 9 | 2181.85 | 3.19 | 0.05 |
| Temp-Autumn | 8 | 2182.43 | 3.77 | 0.04 |
| Temp-Pre + Freeze-Pre | 10 | 2183.21 | 4.55 | 0.02 |
| Base Model | 7 | 2183.78 | 5.12 | 0.02 |

mean pre-breeding temperature varied from $-16.4°C$ to $-6.2°C$ ($-10.7°C \pm 2.3$). The frequency of freeze–thaw events in the autumn ranged from 20 to 57 ($40 \pm 11$) and between 1 and 32 ($15 \pm 7$) in the pre-breeding period.

Brood size ranged from no nestlings (failed nests) to a maximum of five nestlings ($1.7 \pm 1.4$ nestlings). Thirty-five per cent (252 of 718) of nests failed before banding at day 11 and nestling condition varied widely across nests, with residuals from the mass-body size regression ranging from $-18.80$ to $13.44$ ($0.18 \pm 4.1$). Neither mean temperature nor freeze–thaw events showed an increasing trend over the course of our 40-year study (mean temperature: $-0.397 \pm 0.16$, $p > 0.05$, freeze–thaw events: $0.03 \pm 0.025$, $p = 0.02$).

## 3.2. Brood size

Four top models ($\Delta AICc \leq 2$) best predicted brood size (table 1) but each model included the frequency of autumn freeze–thaw events and mean pre-breeding temperature, both of which were negatively correlated with brood size (figure 2a,b; electronic supplementary material, table S3). As expected, female age was positively correlated with brood size (figure 2d; electronic supplementary material, table S3) and lay date showed a strong negative correlation with brood size (figure 2c and table 2; electronic supplementary material, table S3).

## 3.3. Nest success

There was only one top model to predict variation in nest success (whether a nest contained nestlings at the time of banding) and, similar to brood size models, it included both the frequency of autumn freeze–thaw events and mean pre-breeding temperatures (table 3). However, the top model also included an interaction between lay date and the frequency of autumn freeze–thaw events, suggesting that the negative effect of freeze–thaw events on nest success was more pronounced for late nests compared to early nests (estimate and 95% confidence interval; $-0.26$ ($-0.275$– $-0.225$); figure 3a). There was a negative correlation between mean pre-breeding temperatures and nest success ($-0.22$ ($-0.255$– $-0.185$); figure 3b). As expected, female age was positively correlated with nest success ($0.24$ ($-0.22$–$0.7$); figure 3c).

## 3.4. Nestling condition

The top model to explain variation in nestling condition only included climate variables from the autumn: the frequency of freeze–thaw events and mean temperature. Consistent with both the brood

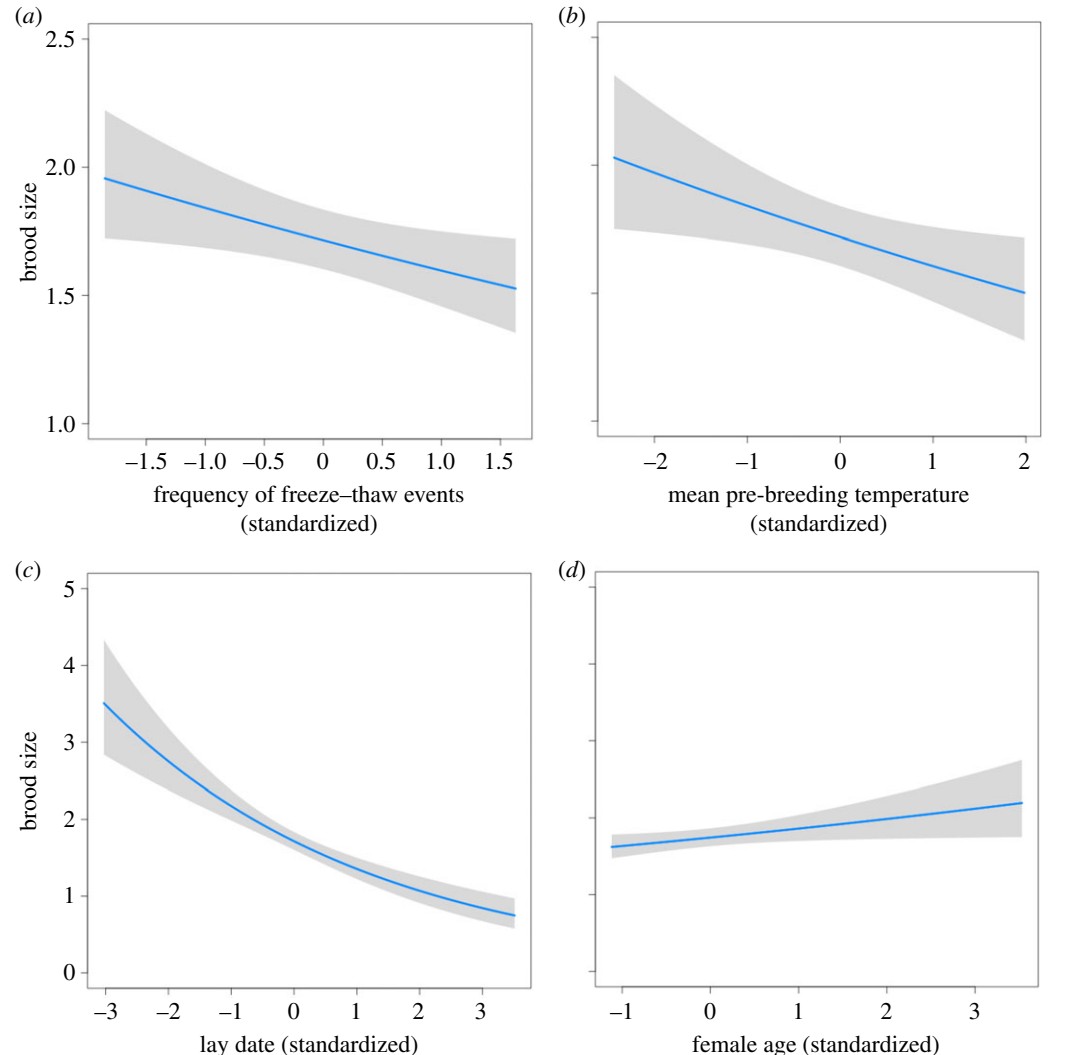

**Figure 2.** Climatic and non-climatic predictors of brood size in Canada jays. The frequency of freeze–thaw events (*a*), mean pre-breeding temperature (*b*) and lay date (*c*) were all negatively correlated with brood size. Female age was positively correlated with brood size (*d*). The shaded area on each graph represents a 95% confidence interval. Each line is taken from the top model as determined through model selection using AICc.

size and nest success models, the frequency of autumn freeze–thaw events was negatively correlated with nestling condition ($-0.70$ ($-1.08$–$-0.33$); figure 4*a*). Mean autumn temperature, however, was positively correlated with nestling condition (0.32 (0.091–0.54); figure 4*b*). As expected, there was evidence for a negative correlation between lay date and nestling condition ($-0.47$ ($-0.77$–$-0.17$); figure 4*c*). Nestling condition was also negatively correlated with brood size, suggesting that lower nestling condition was also associated with a higher number of siblings in the nest ($-0.74$ ($-1.08$–$-0.41$); figure 4*d*).

## 4. Discussion

Our study provides evidence that the number of freeze–thaw events during the autumn caching period and, to a lesser extent, mean temperature in the winter pre-breeding period carry over to influence several measures of reproductive performance in Canada jays during their subsequent late-winter breeding season. Our estimate of the effect size of autumn freeze–thaw events on brood size is similar to the estimated effect size of autumn temperature on brood size in a previous study [17]. However, our work also demonstrates negative effects of autumn freeze–thaw events on both nest success and nestling condition and shows either no effect or a positive effect of autumn temperatures on these same reproductive metrics. Thus, our results support the general notion that cached food could be

**Table 2.** Model comparison to explain variation in nest success of Canada jays using Akaike's information criterion for small sample sizes (AICc). The base model included level of supplementation, male age, female age and lay date but no climate variables. All other models also included these fixed effects. Climate variables added in each model are listed in addition to the total number of parameters in a model ($K$), AICc and $\Delta$AICc scores and AICc weight. Freeze = frequency of freeze–thaw events, Temp = mean temperature, Autumn = autumn caching period (October–November), Pre = Pre-breeding period (January–February).

| model predicting nest success | $K$ | AICc | $\Delta$AICc | AICc weight |
|---|---|---|---|---|
| Freeze-Autumn + Temp-Pre + Freeze-Autumn $\times$ Lay Date | 11 | 767.34 | 0 | 0.58 |
| Freeze-Autumn + Temp-Pre | 10 | 770.24 | 2.90 | 0.15 |
| Freeze-Autumn + Temp-Autumn + Temp-Pre | 11 | 771.73 | 4.39 | 0.07 |
| Freeze-Autumn + Temp-Pre + Freeze-Pre | 11 | 772.26 | 4.92 | 0.05 |
| Temp-Pre | 9 | 773.29 | 5.95 | 0.03 |
| Freeze-Autumn | 9 | 773.55 | 6.21 | 0.03 |
| Freeze-Autumn + Temp-Autumn + Temp-Pre + Freeze-Pre | 12 | 773.71 | 6.37 | 0.03 |
| Freeze-Autumn + Temp-Autumn | 10 | 773.87 | 6.60 | 0.02 |
| Freeze-Pre | 9 | 774.61 | 7.27 | 0.02 |
| Temp-Pre + Freeze-Pre | 10 | 775.25 | 7.91 | 0.01 |
| Base Model | 8 | 775.90 | 8.56 | <0.01 |

**Table 3.** Model comparison to explain variation in nestling condition of Canada jays using Akaike's information criterion for small sample sizes (AICc). The base model included level of supplementation, male age, female age and lay date but no climate variables. All other models also included these fixed effects. Climate variables added in each model are listed in addition to the total number of parameters in a model ($K$), AICc and $\Delta$AICc scores and AICc weight. Models with $\Delta$AICc scores greater than 20 were excluded from this table. Freeze = frequency of freeze–thaw events, Temp = mean temperature, Autumn = autumn caching period (October–November), Pre = Pre-breeding period (January–February).

| model predicting nestling condition | $K$ | AICc | $\Delta$AICc | AICc weight |
|---|---|---|---|---|
| Freeze-Autumn + Temp-Autumn | 11 | 6624.6 | 0 | 0.65 |
| Freeze-Autumn + Temp-Autumn + Freeze-Autumn $\times$ Lay Date | 12 | 6627.4 | 2.8 | 0.16 |
| Freeze-Autumn + Temp-Autumn + Freeze-Pre | 12 | 6628.9 | 4.3 | 0.08 |
| Freeze-Autumn + Temp-Pre + Temp-Pre | 12 | 6629.0 | 4.4 | 0.07 |
| Freeze-Autumn | 10 | 6631.6 | 7.0 | 0.02 |
| Freeze-Autumn + Temp-Autumn + Temp-Pre + Freeze-Pre | 13 | 6632.1 | 7.5 | 0.02 |
| Freeze-Autumn + Temp-Pre | 11 | 6634.5 | 9.9 | <0.01 |
| Freeze-Autumn + Temp-Pre + Freeze-Pre | 12 | 6637.7 | 13.1 | <0.01 |
| Base Model | 9 | 6641.3 | 16.7 | <0.01 |
| Temp-Pre | 10 | 6643.3 | 18.7 | <0.01 |
| Temp-Autumn | 10 | 6643.4 | 18.8 | <0.01 |
| Freeze-Pre | 10 | 6646.3 | 19.8 | <0.01 |

degraded by autumn conditions [17], but provides stronger support for freeze–thaw events as the specific climate mechanism acting to influence perishable cached food.

Experimental studies on food consumed by humans have demonstrated consistent negative effects of freeze–thaw events on food quality [19,20,22–24,42], and it is likely that similar effects occur with perishable food items cached by Canada jays. While we provide support for this hypothesis, it is important to note that we have not directly tested the effect of freeze–thaw events on cached food. Further, our results permit few firm conclusions about the relative importance of the mechanisms by which freeze–thaw events are known to degrade food quality. It seems unlikely that the

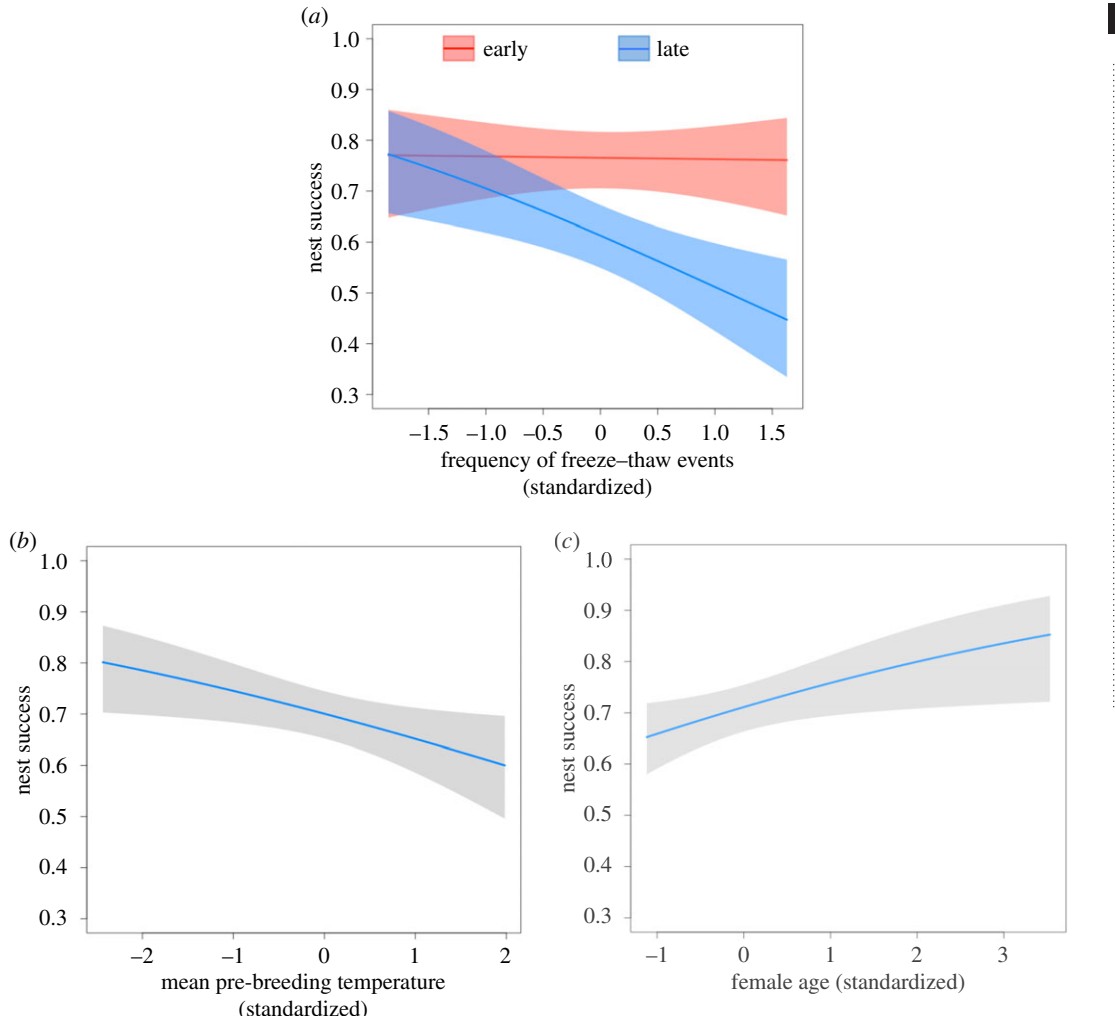

**Figure 3.** Climatic and non-climatic predictors of nest success in Canada jays. (*a*) There was a significant interaction between lay date and the frequency of autumn freeze–thaw events, whereby early nesting pairs (red) were less influenced by freeze–thaw events than late nesting pairs (blue). (*b*) Mean pre-breeding temperatures were negatively correlated with nest success. (*c*) Female age was positively correlated with nest success. The shaded area around each line represents a 95% confidence interval. Each line is taken from the top model as determined through model selection using AICc.

denaturation of proteins alone could be responsible for a majority of the degradation of food caches because protein within a food item is probably not being degraded to the point where it is no longer usable by Canada jays [20,25]. However, it is difficult to differentiate between the degradation caused by the two remaining processes associated with freeze–thaw events. Both drip loss and enhanced microbial degradation during thaw periods stem from the prior physical damage to cell membranes caused by freezing and thawing [20,22,23]. The primary differences between these processes are that degradation due to drip-loss could, in principle, occur in a completely sterile environment and would be dependent on the duration of post-freezing thaws (i.e. longer thaws allow more liquid to leave the cell), whereas food quality degradation attributable to enhanced microbial growth will depend on the intensity (warmth), as well as the duration of post-freezing thaws. These distinctions aside, our finding of an important effect of freeze–thaw events in the autumn, as opposed to those occurring in the pre-breeding period, is equally consistent with degradation due to both drip loss and microbial growth. An autumn freeze–thaw event is expected to have a more negative impact on subsequent reproductive performance simply because the total duration of subsequent thaws, between storage and the eventual breeding-season consumption of a cached food item, will be longer after an autumn freezing event than after a late-winter, pre-breeding event.

Our finding that the total number of autumn freeze–thaw events is negatively correlated with reproductive performance also suggests that continued freezing and thawing is an important determinant of food quality. Above we outlined the processes by which a single freeze–thaw event may

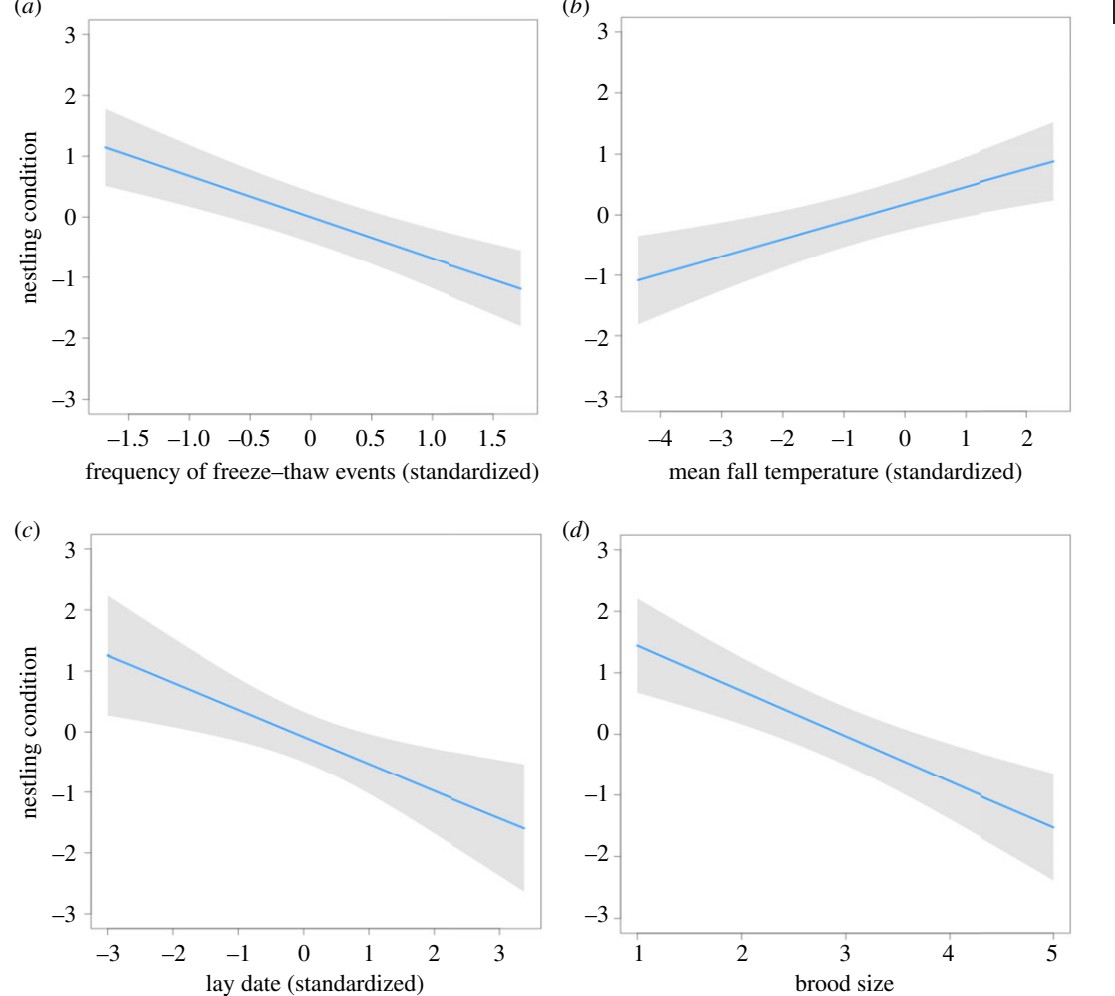

**Figure 4.** Climatic and non-climatic predictors of nestling condition in Canada jays. Nestling condition of Canada jays was negatively correlated with the frequency of autumn freeze−thaw events (a), lay date (c) and brood size (d). Nestling condition was positively correlated with mean autumn temperature (b). The shaded area around each line represents a 95% confidence interval. Each line is taken from the top model as determined through model selection using AICc.

influence food quality, but there is evidence that the negative effects become compounded with each freeze−thaw events that occurs [20,21]. Previous experiments have demonstrated this over the course of relatively few cycles (e.g. 4−6), but it remains to be seen if the negative effects of freeze−thaw events would continue to accumulate over a typical autumn/late winter period in APP, or if there is a point where food quality deteriorates to a point where it is no longer influenced by additional freeze−thaw events.

It is interesting to note that autumn freeze−thaw events appear to be carrying over to influence decisions and outcomes at multiple stages of the breeding period. Both brood size and nest success are directly related to decisions that a breeding pair makes, in particular, how much energy to allocate to reproduction instead of maintaining individual condition [13]. Cached food is used to support pre-breeding weight gain in females [15] and, therefore, if food has degraded prior to clutch initiation, females may choose to invest less energy towards the number of eggs laid and divert it instead towards maintaining their over-winter condition. This is supported by previous work on Canada jays that has demonstrated an effect of autumn climatic conditions on reproduction [17], but not on over-winter survival [13]. Similarly, nest success could be influenced by food available to a female because this would dictate the amount of energy that can be invested in reproduction and when a female should abandon a nest to devote more energy to survival. Although we are unable to identify the cause of failure for most nests, decreased food quality could lead to a greater number of foraging trips by a female or a greater proportion of time spent away from a nest. Both of these scenarios could lead to increased rates of predation because increased activity at the nest could attract predators and time spent away from the nest could mean that females are less able to defend their nest. In contrast to

both brood size and nest success, nestling condition is directly influenced by the quality of food that is being fed to young as they develop [43]. Therefore, when food is degraded by environmental conditions, less energy can be derived from each cached food item and nestling condition would be expected to suffer.

Our analysis also points to an intriguing interaction between events in two different periods of the annual cycle. In terms of nest success, Canada jays nesting later in the season, perhaps due to inexperience or lower food reserves, appear to be more susceptible to variation in the frequency of autumn freeze–thaw events compared to individuals nesting early in the season. Individuals nesting later in the season may be raising nestlings at times when temperatures have warmed sufficiently to degrade cached food stores and when there is still little fresh food available in the environment. Thus, late nesting pairs could be more heavily impacted by poor autumn conditions, while early nesting pairs may still have sufficient cached food stores to raise young, even under poor autumn conditions. The importance of early nesting for reproductive performance has been reported previously [30,35], and, collectively, our work highlights an interesting, potentially opposing effect of long-term changes in climate: warmer autumns, which could lead to an increased number of freeze–thaw events, appear to have a negative effect on reproductive performance [17], whereas warmer late winters appear to have a positive effect because a higher proportion of birds nest earlier in the season and, therefore, have higher reproductive success. How these opposing effects of climate may influence long-term population trends remains to be investigated.

In addition to the observed effects on reproductive performance, climate variables in the autumn and pre-breeding periods could carry over to influence other vital rates, such as juvenile survival. Nestling condition is a key determinant of post-fledging survival and, in many passerine species including European blackbirds (*Turdus merula*), great tits (*Parus major*) and coal tits (*Parus ater*), there is a positive relationship between nestling condition and survival [44–46]. The negative relationship we observed between autumn climate and nestling condition suggests that conditions during the autumn caching period could influence more than just fecundity. Future studies are needed to quantify potential effects of climatic phenomena across critical life stages, which could have a significant downstream impact on population growth rate.

Contrary to the predictions of the hoard-rot hypothesis, we found a positive correlation between autumn temperature and nestling condition. This surprising effect is challenging to explain but one possibility is that warmer temperatures could have a positive influence on the over-winter survival of prey items that become available in the spring during the nestling period. Warmer autumn conditions generally translate to more benign over-wintering conditions that increase arthropod survival [47]. Higher autumn temperatures could, therefore, produce larger or earlier emergence events in the spring and more opportunity for adults to collect greater volumes of food to feed nestlings.

Although we provide evidence that Canada jay reproductive performance is influenced by climate during the autumn and, to a lesser extent, the winter pre-breeding period, we do not yet know if these factors are driving the observed decline of Canada jays in APP. Neither freeze–thaw events nor mean temperature in either the autumn or pre-breeding period showed a significant linear increase over time. Previous studies [16,17] found an increase in mean autumn temperature over time in Algonquin Provincial Park, but immediately after these studies were published a period of cold winters led to the observed trend becoming not significant over time. Thus, while variation in autumn and winter climate may play important roles in determining fecundity, it is possible that fecundity may not be the primary vital rate driving population growth. Consistent with this, observed declines in fecundity in our study population do not match steeper declines in population abundance. There is also no evidence for a corresponding decline in adult survival over the same time period [13]. The primary demographic vital rate driving declines may be juvenile (first year) survival and subsequent lack of recruitment into the population. What is required is an understanding of how environmental factors throughout the annual cycle influence all vital rates, as well as the relative contribution of those vital rates to population growth [48].

Identifying mechanisms that link climate change and fitness are important in order to predict future population responses to climate change [6]. Our results suggest that Canada jay populations in areas experiencing increases in the number of freeze–thaw events and warmer winter temperatures could be particularly susceptible to climate change. Environments at both high elevations and northern latitudes, which include much of the Canada jay's range, are predicted to experience more pronounced changes in temperature and precipitation over time than other regions in North America [49]. These predicted changes in climate across North America could lead to widespread population declines that may not be restricted to the current southern edge of the Canada jay's range.

Our study presents evidence of freeze–thaw events influencing reproductive performance in a single food-caching species, but several other food-caching species could be similarly affected by carry-over effects of environmental phenomena. Following the framework proposed by Sutton *et al.* [9], species that store perishable foods for long periods of time, similar to Canada jays, are predicted to be most susceptible to environmental conditions that influence food quality over time. One species in this high-risk category is the wolverine (*Gulo gulo*), which is declining at the southern edge of its range [50]. However, few other species in this category have sufficient demographic data necessary to evaluate how degradation of cached food could influence either survival or reproduction. Long-term demographic studies are needed to assess how climatic events throughout the year, and in particular the events that occur between storage and retrieval of a food item, could influence either survival or reproduction in these species.

Ethics. Animal care approval was received from the University of Guelph animal care committee and from Canadian Wildlife Services, which approved all marking of individuals through permits to D.S., N.E.F., D.R.N. and A.O.S. Permission to carry out this study in Algonquin Provincial Park was provided by Ontario Parks.

Data accessibility. Data available from the Dryad Digital Repository at: https://doi.org/10.5061/dryad.qd6gr5v [51].

Authors' contributions. All authors contributed to data collection and contributed to the drafting of the manuscript; A.O.S., D.R.N. and D.S. conceived of the study and A.O.S. carried out the statistical analyses. All authors gave final approval for publication.

Competing interests. We have no competing interests.

Funding. This project received financial support from the Natural Sciences and Engineering Research Council of Canada (Discovery Grants to D.R.N. and A.E.M.N., CGS-D Scholarship to N.E.F.), a University of Guelph Research Chair (D.R.N.), Ontario Graduate Scholarship (A.O.S. and N.E.F.), the Ontario Ministry of Natural Resources and Forestry, Ontario Parks and a W. Garfield Weston Foundation Fellowship for Northern Conservation from the Wildlife Conservation Society (A.O.S.).

Acknowledgements. We thank the many volunteers who assisted with nest searching and fieldwork over the course of this study, in particular R. and D. Tozer, G. and D. Hanes, R. Hawkins and the late M. Pageot. This work received additional logistical support from Algonquin Park staff, and S. Dobbyn and P. Gelok from the Ontario Ministry of Natural Resources. We would also like to thank two anonymous reviewers for helpful comments that improved the manuscript.

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
