## [Reviewer comments · Royal Society Open Science]

Review History

RSOS-181754.R0 (Original submission)

Review form: Reviewer 1

Is the manuscript scientifically sound in its present form?

Yes

Are the interpretations and conclusions justified by the results?

Yes

Is the language acceptable?

Yes

Is it clear how to access all supporting data?

Yes

Do you have any ethical concerns with this paper?

No

Have you any concerns about statistical analyses in this paper?

No

Recommendation?

Accept with minor revision (please list in comments)

Comments to the Author(s)

To Author:

The authors have generally done a satisfactory job of responding to comments, but further details are still needed. In general, I think the manuscript would benefit from a more explicit documentation of its limitations regarding measures of reproductive success. First, the authors stated in their response that “we are not able to estimate clutch size”. However, in two previous papers that they cite and that some of the present authors were a part of, clutch size is estimated in the same study system. Second, the use of number of nestlings on day 11 as a measure of nest success is not ideal given that it’s only about halfway through the nestling period. If that’s the best you can do, that’s fine, but the reader should be made aware of this limitation and the implications of this choice should be discussed. Third, more detail is needed on your measure of body condition in relation to day of banding. Please see detailed comments below.

Lines 56-58. You still haven’t compared the strength of your effects to those found by Waite and Strickland. How much stronger and more convincing are your effects? Also, I’m not entirely sure it’s fair to say there was no evidence that fall temperature affected brood size in that study. They found a negative trend of similar strength to clutch size at $P = 0.08$.

Lines 96-99. I still have some issues here. How long is the nestling period for Canada Jays? I’m seeing estimates as long as 22-24 days. This means that you’re estimating nest success only halfway through the nestling period. Do you have any evidence that nestling presence at day 11 is indicative of overall nest success at the time of fledging? From your response to previous comments it appears not. The limitations of your definition of “success” needs to be made really explicit to the reader. I think you need to state the length of the nesting period, why you did not check nests after day 11, and argue to the best of your ability why your measure of nest success is still likely to be accurate.

Lines 107-116. Do you have evidence that this is a good measure of condition (i.e., that it has documented consequences)? If not, that’s ok as it’s a standard assumption - but you should state that you’re making an assumption that this measures body condition.

Lines 105-106. Does excluding failed nests reduce sample size mentioned in line 96? Or is that still accurate? Should be made clear for the reader either way.

Line 191. But above you said that “brood size” excluded failed nests. Here they are included. Clarify. I think it makes the most sense to state how many nests failed in the methods, state that you excluded failed nests, and then only report on the nests in the analysis in the results section

Line 192. Since you have no information about why nests fail, I think you should state that explicitly. I think most readers are going to immediately want to know why nests are failing.

Line 192-194. Does any of this variation stem from the variation in measuring day? Looking back to Derbyshire et al. 2015 it looks like nestlings were measured anywhere from 7 to 12 days, a potentially big difference for fast-growing nestlings. Was day of measurement controlled for in your body condition measure? If not, why not?

Line 267 – 270. In your response to previous comments you state that you were not able to estimate clutch size. However, in Derbyshire et al. 2015 clutch size was able to be estimated for 325 of 394 nests from presumably the same data set. Why the discrepancy? If for some reason you can't estimate clutch size, then you need to make it explicit that you're assuming brood size at day 11 is representative of clutch size since the mechanism you're mentioning here is really focused on number of eggs laid.

Lines 272 – 274. As I mentioned in a previous review, the fact that you haven't given any information about how nests fail in this species makes it hard to evaluate your argument here. Abandonment is just one way a nest could fail. What about predation, starvation, etc? I realize are not able to determine why a nest failed, but nonetheless it seems important to understand the mechanism and must be discussed. Is it possible that a food limited female might have a tradeoff between time spent foraging away from the nest and time defending the nest from predators? In this manner could food limitation decrease nest success indirectly through increased predation? The reader needs to understand that you cannot determine why nests fail, how that limits your understanding of the precise mechanisms behind the effects you've found evidence for, and what future research is needed to clarify.

Lines 313-316. In your response to a previous comment you state that "Immediately after these studies were published, we had a period of cold winters that have caused the observed trend to become non-significant over time." Ok – that makes sense, but probably should be explained so the reader is clear on why there used to be a trend, but isn't anymore. Moreover, if you're doing an analysis, shouldn't it be in the results?

Lines 326-328. You state in your response that you don't have any evidence that such areas where increased in temperature and freeze-thaw event exist. This statement you make in these lines implies that these areas definitely exist – edit text for clarification. Maybe you could be more clear about the "pronounced climactic changes" to strengthen this ooverall argument as well?

Review form: Reviewer 2

Is the manuscript scientifically sound in its present form?

Yes

Are the interpretations and conclusions justified by the results?

Yes

Is the language acceptable?

Yes

Is it clear how to access all supporting data?

Yes

Do you have any ethical concerns with this paper?

No

Have you any concerns about statistical analyses in this paper?

No

Recommendation?

Accept as is

Comments to the Author(s)

It is the second time that I am reviewing the paper "Fall freeze-thaw events carry over to depress late-winter reproductive performance in Canada jays" by Sutton and colleagues. The authors have addressed most of my comments in the revised version of the manuscript and I congratulate them for their great work and this very interesting study.

Decision letter (RSOS-181754.R0)

22-Feb-2019

Dear Dr Sutton

On behalf of the Editors, I am pleased to inform you that your Manuscript RSOS-181754 entitled "Fall freeze-thaw events carry over to depress late-winter reproductive performance in Canada Jays" has been accepted for publication in Royal Society Open Science subject to minor revision in accordance with the referee suggestions. Please find the referees' comments at the end of this email.

The reviewers and handling editors have recommended publication, but also suggest some minor revisions to your manuscript. Therefore, I invite you to respond to the comments and revise your manuscript.

- Ethics statement

- Data accessibility

<http://datadryad.org/submit?journalID=RSOS&manu=RSOS-181754>

- Competing interests

- Authors' contributions

- Acknowledgements

- Funding statement

Because the schedule for publication is very tight, it is a condition of publication that you submit the revised version of your manuscript before 03-Mar-2019. Please note that the revision deadline will expire at 00.00am on this date. If you do not think you will be able to meet this date please let me know immediately.

on behalf of Dr Ryan Earley (Associate Editor) and Professor Kevin Padian (Subject Editor)
openscience@royalsociety.org

Reviewer comments to Author:

Reviewer: 1

Comments to the Author(s)

To Author:

The authors have generally done a satisfactory job of responding to comments, but further details are still needed. In general, I think the manuscript would benefit from a more explicit documentation of its limitations regarding measures of reproductive success. First, the authors stated in their response that “we are not able to estimate clutch size”. However, in two previous papers that they cite and that some of the present authors were a part of, clutch size is estimated in the same study system. Second, the use of number of nestlings on day 11 as a measure of nest success is not ideal given that it’s only about halfway through the nestling period. If that’s the best you can do, that’s fine, but the reader should be made aware of this limitation and the implications of this choice should be discussed. Third, more detail is needed on your measure of body condition in relation to day of banding. Please see detailed comments below.

Lines 56-58. You still haven’t compared the strength of your effects to those found by Waite and Strickland. How much stronger and more convincing are your effects? Also, I’m not entirely sure it’s fair to say there was no evidence that fall temperature affected brood size in that study. They found a negative trend of similar strength to clutch size at $P = 0.08$.

Lines 96-99. I still have some issues here. How long is the nestling period for Canada Jays? I’m seeing estimates as long as 22-24 days. This means that you’re estimating nest success only halfway through the nestling period. Do you have any evidence that nestling presence at day 11 is indicative of overall nest success at the time of fledging? From your response to previous comments it appears not. The limitations of your definition of “success” needs to be made really explicit to the reader. I think you need to state the length of the nesting period, why you did not check nests after day 11, and argue to the best of your ability why your measure of nest success is still likely to be accurate.

Lines 107-116. Do you have evidence that this is a good measure of condition (i.e., that it has documented consequences)? If not, that’s ok as it’s a standard assumption - but you should state that you’re making an assumption that this measures body condition.

Lines 105-106. Does excluding failed nests reduce sample size mentioned in line 96? Or is that still accurate? Should be made clear for the reader either way.

Line 191. But above you said that “brood size” excluded failed nests. Here they are included. Clarify. I think it makes the most sense to state how many nests failed in the methods, state that you excluded failed nests, and then only report on the nests in the analysis in the results section

Line 192. Since you have no information about why nests fail, I think you should state that explicitly. I think most readers are going to immediately want to know why nests are failing.

Line 192-194. Does any of this variation stem from the variation in measuring day? Looking back to Derbyshire et al. 2015 it looks like nestlings were measured anywhere from 7 to 12 days, a potentially big difference for fast-growing nestlings. Was day of measurement controlled for in your body condition measure? If not, why not?

Line 267 – 270. In your response to previous comments you state that you were not able to estimate clutch size. However, in Derbyshire et al. 2015 clutch size was able to be estimated for 325 of 394 nests from presumably the same data set. Why the discrepancy? If for some reason you

can't estimate clutch size, then you need to make it explicit that you're assuming brood size at day 11 is representative of clutch size since the mechanism you're mentioning here is really focused on number of eggs laid.

Lines 272 – 274. As I mentioned in a previous review, the fact that you haven't given any information about how nests fail in this species makes it hard to evaluate your argument here. Abandonment is just one way a nest could fail. What about predation, starvation, etc? I realize are not able to determine why a nest failed, but nonetheless it seems important to understand the mechanism and must be discussed. Is it possible that a food limited female might have a tradeoff between time spent foraging away from the nest and time defending the nest from predators? In this manner could food limitation decrease nest success indirectly through increased predation? The reader needs to understand that you cannot determine why nests fail, how that limits your understanding of the precise mechanisms behind the effects you've found evidence for, and what future research is needed to clarify.

Lines 313-316. In your response to a previous comment you state that "Immediately after these studies were published, we had a period of cold winters that have caused the observed trend to become non-significant over time." Ok – that makes sense, but probably should be explained so the reader is clear on why there used to be a trend, but isn't anymore. Moreover, if you're doing an analysis, shouldn't it be in the results?

Lines 326-328. You state in your response that you don't have any evidence that such areas where increased in temperature and freeze-thaw event exist. This statement you make in these lines implies that these areas definitely exist – edit text for clarification. Maybe you could be more clear about the "pronounced climactic changes" to strengthen this ooverall argument as well?

Reviewer: 2

Comments to the Author(s)

It is the second time that I am reviewing the paper "Fall freeze-thaw events carry over to depress late-winter reproductive performance in Canada jays" by Sutton and colleagues. The authors have addressed most of my comments in the revised version of the manuscript and I congratulate them for their great work and this very interesting study.

Author's Response to Decision Letter for (RSOS-181754.R0)

See Appendix A.

Decision letter (RSOS-181754.R1)

07-Mar-2019

Dear Dr Sutton,

I am pleased to inform you that your manuscript entitled "Fall freeze-thaw events carry over to depress late-winter reproductive performance in Canada Jays" is now accepted for publication in Royal Society Open Science.

on behalf of Dr Ryan Earley (Associate Editor) and Kevin Padian (Subject Editor)
openscience@royalsociety.org

Appendix A

All our responses to reviewers appear below their comments and are bolded for clarity. Additionally, all revisions included in the text of our manuscript appear in red.

Reviewer comments to Author:

Reviewer: 1

Comments to the Author(s)

To Author:

The authors have generally done a satisfactory job of responding to comments, but further details are still needed. In general, I think the manuscript would benefit from a more explicit documentation of its limitations regarding measures of reproductive success. First, the authors stated in their response that "we are not able to estimate clutch size". However, in two previous papers that they cite and that some of the present authors were a part of, clutch size is estimated in the same study system. Second, the use of number of nestlings on day 11 as a measure of nest success is not ideal given that it's only about halfway through the nestling period. If that's the best you can do, that's fine, but the reader should be made aware of this limitation and the implications of this choice should be discussed. Third, more detail is needed on your measure of body condition in relation to day of banding. Please see detailed comments below.

Lines 56-58. You still haven't compared the strength of your effects to those found by Waite and Strickland. How much stronger and more convincing are your effects? Also, I'm not entirely sure it's fair to say there was no evidence that fall temperature affected brood size in that study. They found a negative trend of similar strength to clutch size at $P = 0.08$.

As suggested, we have compared the strength of the effects in our study to those presented in Waite and Strickland (2006) in the discussion (lines 243-247).

Lines 96-99. I still have some issues here. How long is the nestling period for Canada Jays? I'm seeing estimates as long as 22-24 days. This means that you're estimating nest success only halfway through the nestling period. Do you have any evidence that nestling presence at day 11 is indicative of overall nest success at the time of fledging? From your response to previous comments it appears not. The limitations of your definition of "success" needs to be made really explicit to the reader. I think you need to state the length of the

nesting period, why you did not check nests after day 11, and argue to the best of your ability why your measure of nest success is still likely to be accurate.

As suggested, we have included the length of the nesting period in the manuscript (lines 98-99). We have also further expressed the limitations of our nest success estimate and why we think it is still a good indicator of success (lines 113-118).

More generally, we do have a small sample size of nests that were monitored until fledge. In most of these cases, if a nest was successful at the time of banding it did successfully fledge young. Historically, nests were not monitored to determine if young successfully fledged due to logistical constraints associated with accessing nests, counting nestlings still in the nest and locating young post fledge.

Lines 107-116. Do you have evidence that this is a good measure of condition (i.e., that it has documented consequences)? If not, that's ok as it's a standard assumption - but you should state that you're making an assumption that this measures body condition.

As suggested, we have stated that we assume our estimate of condition is a good measure of body condition (lines 128-130).

Lines 105-106. Does excluding failed nests reduce sample size mentioned in line 96? Or is that still accurate? Should be made clear for the reader either way.

As suggested, we have clarified that the stated mean and range of nests monitored each year does include failed nests (lines 96-97).

Line 191. But above you said that "brood size" excluded failed nests. Here they are included. Clarify. I think it makes the most sense to state how many nests failed in the methods, state that you excluded failed nests, and then only report on the nests in the analysis in the results section

As suggested, we have clarified that in subsequent analyses of brood size that all failed nests were excluded (lines 168-170). The stated range of brood size in this section of the results is a summary of brood size across all nests.

Line 192. Since you have no information about why nests fail, I think you should state that explicitly. I think most readers are going to immediately want to know why nests are failing.

As suggested, we have clarified that we do not know the cause of failure of nests in the discussion (lines 293-297).

Line 192-194. Does any of this variation stem from the variation in measuring day? Looking back to Derbyshire et al. 2015 it looks like nestlings were measured anywhere from 7 to 12 days, a potentially big difference for fast-growing nestlings. Was day of measurement controlled for in your body condition measure? If not, why not?

In Derbyshire et al. (2015), a negative exponential growth curve was fit to the data based on known age nestlings. This curve represented the best relationship between size and mass of growing nestlings. This relationship is based solely on the size and mass of an individual and therefore variation in the age at banding does not drive variation in our condition estimates because size is controlled for.

Line 267 – 270. In your response to previous comments you state that you were not able to estimate clutch size. However, in Derbyshire et al. 2015 clutch size was able to be estimated for 325 of 394 nests from presumably the same data set. Why the discrepancy? If for some reason you can't estimate clutch size, then you need to make it explicit that you're assuming brood size at day 11 is representative of clutch size since the mechanism you're mentioning here is really focused on number of eggs laid.

As suggested, we have edited the text in the methods section (lines 108-112) to reflect this uncertainty.

We cannot estimate clutch size independent of brood size because we do not check nest contents before banding. At the time of banding, there may be eggs in some nests, which suggests that clutch size was larger than the observed brood size. This is a relatively infrequent occurrence and therefore we assume that clutch size and brood size are the same for most nests.

Lines 272 – 274. As I mentioned in a previous review, the fact that you haven't given any information about how nests fail in this species makes it hard to evaluate your argument here. Abandonment is just one way a nest could fail. What about predation, starvation, etc? I

realize are not able to determine why a nest failed, but nonetheless it seems important to understand the mechanism and must be discussed. Is it possible that a food limited female might have a tradeoff between time spent foraging away from the nest and time defending the nest from predators? In this manner could food limitation decrease nest success indirectly through increased predation? The reader needs to understand that you cannot determine why nests fail, how that limits your understanding of the precise mechanisms behind the effects you've found evidence for, and what future research is needed to clarify.

As suggested, we have explicitly stated our uncertainty as to why nests fail and have also discussed how decreased food quality could affect predation rates in addition to why breeding pairs may abandon nests as they experience decreasing food quality (line 293-297).

Lines 313-316. In your response to a previous comment you state that "Immediately after these studies were published, we had a period of cold winters that have caused the observed trend to become non-significant over time." Ok – that makes sense, but probably should be explained so the reader is clear on why there used to be a trend, but isn't anymore. Moreover, if you're doing an analysis, shouldn't it be in the results?

As suggested, we have indicated in the discussion why our reported trend in mean fall temperature differs from previous studies (lines 338-340). We have also moved the presentation of the statistical results of our linear regression to the results section (lines 209-211).

Lines 326-328. You state in your response that you don't have any evidence that such areas where increased in temperature and freeze-thaw event exist. This statement you make in these lines implies that these areas definitely exist – edit text for clarification. Maybe you could be more clear about the "pronounced climactic changes" to strengthen this overall argument as well?

As suggested, we have edited the text for clarification (line 354).

Reviewer: 2

Comments to the Author(s)

It is the second time that I am reviewing the paper "Fall freeze-thaw events carry over to depress late-winter reproductive performance in

Canada jays" by Sutton and colleagues. The authors have addressed most of my comments in the revised version of the manuscript and I congratulate them for their great work and this very interesting study.